

# Soil fertility along toposequences of the East India Plateau and implications for productivity and sustainability

Peter S Cornish[1], Ashok Kumar[2,3] and Sudipta Das[2,4]

[1]University of Western Sydney, Hawkesbury Campus, Locked Bag 1797, Penrith NSW Australia
[2]Professional Assistance for Development Action, Purulia (PRADAN), West Bengal, India
[3]Present address: Transforming Rural India (TRIF), Ranchi, Jharkhand, India
[4]Present address:, Collective for Integrated Livelihood Initiatives (CInI), Deoghar, Jharkhand, India

*Correspondence to*: Peter Cornish (p.cornish@westernsydney.edu.au)

**Abstract.** In common with other undulating landscapes in Asia, rice (*Oryza sativa*) on the East India Plateau (EIP) was once confined to hydrologic discharge areas or 'lowlands', but progressive terracing has now allowed rainfed transplanted rice to encroach upon 'upland' recharge areas, with potential effects on both hydrology and soil fertility. Hydrologic variation down the toposequence and its implications for rice production have been well documented, but not the variation in soil fertility. Measurements of surface-soil fertility in seven of 24 EIP Districts were used to evaluate variation between and within small watersheds stratified down the toposequence into six land classes that reflect hydrology and land use (three with rice and three without, 36 fields/watershed). We aimed to provide a basis for future research to improve soil fertility management. Soils overall were acid, with 14% of fields requiring liming (pH <5.0) and 44% requiring further acidification to be managed (pH 5.0-5.4). Organic carbon (OC, mean 0.9%) and cation exchange capacity (CEC, mean 10.7 (cmolc/kg) were low. Available phosphorus (P) was mostly very low (mean Bray-P 4.3 mg kg$^{-1}$) and extractable potassium (K) low to marginal (mean 88 mg kg$^{-1}$). Non-rice soils generally had lower pH, OC and CEC than rice soils, but higher P and K. Amongst rice fields, those higher in the toposequence had lower pH, OC and CEC but more P and K. These results are discussed in the context of nutrient flows in the landscape, leading to the conclusion that terracing uplands has reduced the delivery of sediment-bound P to lowlands where, even with organic P recycling, low inputs of inorganic fertiliser have led soil P to decline and become the primary constraint to yield. Soil K is on the same trajectory. Fertiliser-use must increase substantially to sustain the system, a requirement that will challenge the risk-averse subsistence farmers. Field-specific fertiliser and lime recommendations are needed despite systematic toposequence differences, because of variability between fields within land classes.

## 1. Introduction

Much of the rainfed transplanted, or 'lowland', rice (*Oryza sativa*) in South and Southeast Asia is grown in toposequences where relatively small differences in elevation can lead to differentiation in hydrological conditions, soil properties, and often yield (Homma et al., 2003; Fuwa et al., 2007; Tsubo et al., 2007; Boling et al., 2008; Cornish et al., 2010, 2015a). These toposequence studies agree on the hydrologic changes down toposequences but draw varying conclusions about soil fertility. They included relatively few locations, so site-specific factors may have determined the varying outcomes.





The present larger study of fertility down the toposequence involved seven watersheds on the East India Plateau (EIP), which rises south of the eastern Indo-Gangetic Plain and west of the coastal plain of the Bay of Bengal. The study built on hydrologic measurements and agronomic experiments in two case-study watersheds, including water balance modelling that demonstrated

the regional applicability of hydrologic differentiation in the landscape (Cornish et al., 2015a). Geographically broader evaluation of soil chemical fertility is required before extending the agronomic findings of Cornish et al. (2015b) regionally. Transplanted rice on the EIP was once confined to hydrologic discharge areas, or 'lowlands' (Fig. 1). Mid-slopes and uplands are hydrologic recharge areas. Population pressure has led to hillslopes being terraced and bunded for transplanted rice, starting with foot-slopes ('medium-lowland') and progressing to mid-slopes ('medium-upland') that now comprise the main rice area

(Cornish et al., 2015a). Terracing permits lowland rice technology to be used in uplands.

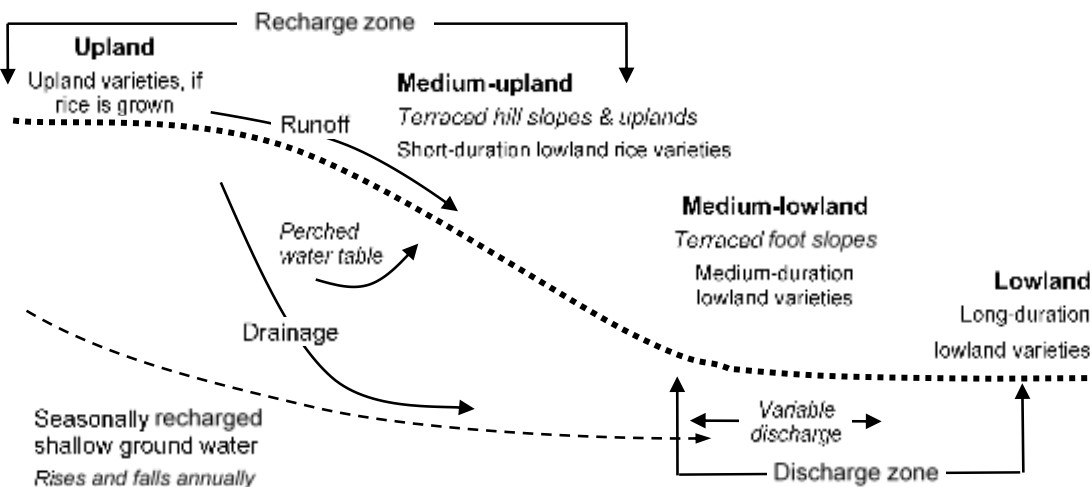

**Fig. 1.** EIP landscape schematic (after Cornish et al., 2015a). Vertical relief is typically 2-100 m and toposequence length 0.5-5 km. Discharge areas may be defined drainage lines, ill-defined low-lying areas, or ephemeral streams. Medium-lowland is a discharge area in wetter years only. Lowlands receive runoff and discharge and historically have been used for lowland rice.

Terraced and bunded foot-slopes and hillslopes comprise around 80% of the present EIP rainfed rice area. Uplands may be categorised into several classes depending on the potential for crops (see text). Upland (non-transplanted) rice is uncommon.

Rice yield improvement on the EIP has been slow, with paddy yields for 2011-2015 averaging <1.9 t ha$^{-1}$ (IRRI, 2019a). This is despite high precipitation (1,100-1,600 mm y$^{-1}$) and good agricultural potential (Sikka et al., 2009). The reasons for low

yield have received relatively little attention, but recent on-farm research and water balance modelling revealed that flooding requirements for transplanted rice are often not met on medium-uplands, leading to frequent yield reductions and recurring crop failure (Cornish et al., 2015a). However, on this same land class, there is adequate soil water in all years for monsoon-season crops that do not require ponding, including dry-bed direct-seeded rice and vegetables (Cornish et al., 2015b). Hence



there is significant potential to raise rice yields and to diversify crops on land unsuited to transplanted rice. These authors
observed that good nutrition would be needed to capture the opportunities provided by rainfall. However, fertiliser inputs are presently low in this region compared with elsewhere in India (FAO, 2005; Anon., 2013).

Soils of the EIP are often said to be acid and infertile (Edmonds et al., 2006; Sikka et al., 2009). These generalisations are based on 1:50000 scale soil maps (http://agri.jharkhand.gov.in/resources/Soil_Inventory/Ranchi_Soil_Analysis.pdf) that sometimes include summaries of field data (Government of Jharkhand, 2019), and a small number of village-scale research
studies (Itfikar et al., 2009; Cornish et al., 2010). Large-scale maps reflect the underlying geology and regional topography, but they do not reflect the local topography or the results of terracing and varying histories of rice culture. Any soil sampling behind maps is grid-based and not designed to identify topographic or land-use effects. Hence, the available maps inadequately reflect the current status of soil fertility in these landscapes. A more nuanced understanding of soil fertility and its variation in the landscape is needed to complement large-scale maps and to inform improved crop nutrient management. For example,
it may be possible to develop fertiliser guidelines based on topographic position as suggested by Boling et al. (2008).

This paper reports a study of soil fertility in six of the 24 Districts in Jharkhand and one in western West Bengal. Earlier research in Purulia District, West Bengal (Cornish et al., 2010) led to a focus on soil pH, organic carbon (OC), cation exchange capacity (CEC), and the macronutrients phosphorus (P) and potassium (K) as indicators of chemical fertility. Our aims were to test the hypothesis that toposequence position affects soil fertility and to provide a basis for future research and extension
to improve management of soil fertility and plant nutrition. We also examined if fertiliser recommendations could be tailored usefully to topographic position.

## 2. Materials and Methods

The East India Plateau (EIP) or Chhota Nagpur Plateau is a series of plateaus, hills and valleys in an undulating landscape with an average elevation ~500 m and occasional higher peaks. It comprises much of the State of Jharkhand and parts of
Chhattisgarh, West Bengal, Bihar and Odisha and is classified as Agroecological Region 12, a hot sub-humid ecoregion with red and lateritic soils (FAO, 2005). The EIP has an area of 65,000 km$^2$ with a population of ~40 million people, 70% of whom are farmers with <1 ha land fragmented into small fields along the toposequence. The region is predominantly low-input, low-output subsistence farming. Monoculture rice is grown in a rice-fallow system although Cornish et al. (2015b) have demonstrated the potential to intensify cropping with a range of crops following rice in the non-rainy season.
The Government of Jharkhand (2019) provides a broad description of the physiography and geology of all Districts with summaries of soil properties based on 1.5 km grid samples. Soil parent materials are primarily igneous (granite) and metamorphic (gneiss, schist) rocks, with colluvium in valley floors. The soils are primarily hyperthermic and classified as moderately developed Alfisols, slightly developed Inceptisols or undeveloped Entisols. Where rice is grown, soils could be classified as 'Anthroposols' because of the cut and fill associated with terracing and the subsequent effects of wet tillage and
ponding on soil profile characteristics.



## 2.1 Site locations

Seven Districts of the EIP were selected to represent variation in underlying geology and to give wide geographic spread across Jharkhand and adjacent West Bengal. These were Godda and Dumka to the north-east of the capital of Jharkhand (Ranchi), Khunti and West Singhbhum to the south, Gumla to the east, and Lohardaga to the north-east of Ranchi. A further location was Bankura in West Bengal, to the east of previous sampling in Purulia District (Cornish et al., 2010). Site selection within these Districts was constrained by security issues, the need for reasonable access, and the desirability of having collaborating farmers to assist with local knowledge, field work and interpretation of results. This effectively meant that sampling was constrained to villages where the NGO PRADAN had a presence. Within each District, a micro-watershed, typically <5 km$^2$, was selected following negotiations with local villagers.

## 2.2 Sampling

A stratified random sample of fields was taken within each watershed based on an assessment of local hydrology made together with villagers who recognise hydrologically distinct land classes and manage them accordingly, e.g. by choosing appropriate rice varieties (Fig. 1). Land classes include lowlands, medium-lowlands, medium-uplands, and uplands that are categorised as (i) land near the homestead to which harvested materials are generally taken (this land may be cropped to vegetables), (ii) arable land near medium-upland that is unsuitable for rice but may be suitable for other rainfed crops, and (iii) non-arable upland that is unsuitable for any crops and is usually grazed as common land. Non-arable upland may be too shallow or stony for crop production, under forest, or degraded by over-grazing and soil erosion. Degraded areas in the non-arable uplands were avoided. Non-degraded non-arable uplands provide an indication of inherent soil fertility.

Fields were sampled in May, 2010 prior to the monsoon. Six fields were selected at random within each of the six land-classes in all seven watersheds, a total 252 fields. Care was taken to avoid fields located near trees or having other obvious anomalies. In each field, soil was sampled along a transect at four locations, from the vertical face of a hole dug to 100 mm depth. These samples were bulked and air-dried for later analysis at the Indian Institute for Soil Science, Bhopal (IISS).

Soil sampling in India is most commonly to 150 mm, but in our experience primary tillage on the EIP is still undertaken with animals, and is rarely deeper than 100 mm. We considered that deeper sampling would risk diluting the nutrient-enriched surface layer, resulting in unreasonably low values.

Unpublished data are also presented for soil profile pH$_w$ from the site of previous studies in Purulia District (Cornish et al., 2010). A total of 54 rice fields along the toposequence were sampled, with a single 100-mm auger hole in the centre of each field in depth increments of 0-10, 10-30, 30-60 and 60-90 cm.

## 2.3 Soil analysis

For quality assurance, a subset of 30 samples was first provided 'blind' to three laboratories for comparative analysis, before settling on IISS, Bhopal.

Organic Carbon (OC) was determined by the Walkley and Black (1934) method, Cation exchange capacity (CEC) by Blakemore et al. (1987) (only 3 of the 6 fields per watershed/land class), and $pH_w$ by glass electrode in a 1:5 water suspension. Plant-available soil P was analysed by the Bray-1 method (Bray and Kurtz, 1945). In India, the use of Bray-1 is confined to

soils with $pH_w$ <7.2. A small number of our samples had $pH_w$ >7.2, but none were calcareous. Exchangeable soil K was estimated by flame photometer after neutral normal ammonium acetate extraction (Spencer and Govaars, 1982) and the values $(cmol_c/kg)$ were multiplied by 390 to arrive at extractable K (mg kg$^{-1}$) (Peverill et al., 2009). Available nitrogen (N) is agronomically important but it was not assessed as it is universally very deficient without N-fertiliser. According to the Government of Jharkhand (2019), Boron (B) and sulphur (S) may be deficient in some soils, but they were not assessed.

The experimental design was a randomised complete block with subsampling (Steele and Torrie, 1980; Tangren, 2002) where watersheds were the blocks, land classes were the treatments and sub samples were the fields within land class. One-way ANOVAs were also undertaken of individual watersheds across land classes and for individual land classes across watersheds. These analyses provided the standard errors (s.e.) in tables of treatment means. Means and s.e. are tabulated without applying statistical comparisons to individual means, following Riley (2001). Analysis of variance used S-Plus 6. Data were

summarised in box plots using Excel in Office Professional Plus (2016) with the median and four quartile values from minimum to maximum shown by horizontal bars, the mean by a 'X', plus outliers (>1.5 x the interquartile range above the upper quartile and below the lower quartile). Separate box plots were constructed for the combined non-rice and combined rice fields. Data for non-rice and rice fields were also compared using t-tests in Excel.

## 3. Results and Discussion

For all analytes, the overall ANOVA revealed significant (P<0.01) main effects for blocks (watershed) and treatments (land class) and a significant (P<0.01) interaction. Watershed differences appeared to reflect differences in underlying geology, resulting in generally higher pH at Godda, lower pH and CEC at Khunti, and higher K at Gumla.

### 3.1 Soil pH

Surface soil $pH_w$ was generally low (Table 1) but not universally so. The variation in $pH_w$ was striking, ranging in non-rice

fields from 4.4 to 6.3 and in rice fields from 4.4 to 7.8 (Fig. 2a). Non-rice upland soils on average were more acid than rice soils (mean $pH_w$ 5.35 v 5.72, P<0.01).

Only 17% of non-rice soils and 10% of rice soils had a $pH_w$ <5.0 that would be classified as strongly acid. Multiple effects on crop growth occur at $pH_w$ <5.0, and remediation with lime should be considered (Peverill et al., 1999). Lowland rice may not require liming because pH shifts towards neutrality with flooding (Seng et al., 2006), but rice land may still need liming if rice

is followed by a non-flooded crop. A further 45% of non-rice fields and 43% of rice fields were in the range $pH_w$ 5.0-5.4, which puts them at risk of crop growth effects from further soil acidification.





Amongst rice soils, there was a trend towards higher $pH_w$ down the toposequence (Table 1), although less so than for the surface soil at Purulia (Cornish et al., 2010). The trend was consistent across watersheds, with the exception of West Singhbhum where medium-upland was inexplicably less acid than elsewhere in the watershed.

**Table 1.** Soil pH in seven EIP watersheds. Values are the mean of six fields.

| Watershed | Rice land | | | Non-rice land | | | Mean | s.e. |
|---|---|---|---|---|---|---|---|---|
| | Lowland | Medium lowland | Medium upland | Arable upland | Non-arable upland | Homestead | | |
| Gumla | 5.18 | 5.32 | 4.95 | 5.22 | 5.49 | 5.64 | 5.30 | 0.09 |
| Lohardaga | 6.13 | 6.73 | 5.53 | 5.65 | 5.38 | 5.55 | 5.83 | 0.09 |
| W Singhbhum | 5.20 | 5.21 | 5.66 | 5.16 | 5.05 | 5.40 | 5.28 | 0.08 |
| Khunti | 5.10 | 4.78 | 5.03 | 4.87 | 4.68 | 4.64 | 4.85 | 0.07 |
| Dumka | 6.02 | 5.88 | 5.44 | 5.59 | 5.02 | 5.19 | 5.52 | 0.07 |
| Godda | 6.52 | 6.65 | 7.21 | 7.37 | 6.20 | 5.85 | 6.63 | 0.07 |
| Bankura | 7.15 | 5.39 | 5.17 | 5.47 | 5.44 | 5.52 | 5.69 | 0.09 |
| Mean | 5.90 | 5.71 | 5.57 | 5.62 | 5.32 | 5.40 | 5.59 | 0.09 |
| s.e. | 0.11 | 0.09 | 0.09 | 0.07 | 0.05 | 0.08 | 0.08 | |

The results for soil profile $pH_w$ in rice fields at Purulia reveal an increase from $pH_w$ 5.6 in surface soil to $pH_w$ 7.1 in subsoil (Fig. 2b). If this result is representative, then any remediation of soil acidity will concern only surface soil.

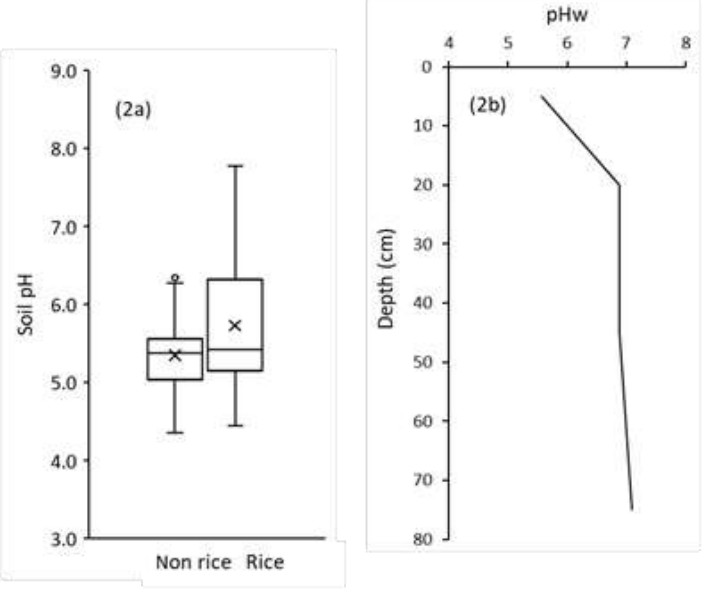

**Fig. 2.** Soil $pH_W$ in (a) surface soil across 7 EIP watersheds and (b) the mean of 54 rice field soil profiles at Purulia District.

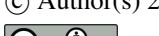


### 3.2 Soil organic carbon

Mean OC in the surface soils of the seven EIP watersheds was 0.9%, with a range among watersheds of 0.5-1.2% (Table 2). These values are towards the lower end of OC reported globally for non-arid areas (FAO and ITPS, 2018). However, they are comparable to soil surveys in the region, which classify them as 'medium' to 'high' in OC (Government of Jharkhand, 2019).

**Table 2.** Organic carbon (%) in seven EIP watersheds.

| Watershed | Rice land | | | Non-rice land | | | Mean | s.e. |
|---|---|---|---|---|---|---|---|---|
| | Lowland | Medium lowland | Medium upland | Arable upland | Non-arable upland | Homestead | | |
| Gumla | 0.78 | 0.72 | 0.72 | 0.59 | 0.75 | 0.72 | 0.71 | 0.04 |
| Lohardaga | 0.61 | 0.61 | 0.44 | 0.52 | 0.48 | 0.42 | 0.51 | NS |
| W Singhbhum | 1.03 | 1.09 | 0.77 | 0.63 | 0.94 | 1.07 | 0.92 | 0.12 |
| Khunti | 1.37 | 1.17 | 0.89 | 0.70 | 0.66 | 0.59 | 0.90 | 0.15 |
| Dumka | 0.89 | 0.76 | 1.03 | 0.95 | 0.62 | 0.82 | 0.84 | 0.08 |
| Godda | 1.10 | 1.31 | 1.13 | 1.37 | 1.17 | 1.00 | 1.18 | NS |
| Bankura | 1.50 | 1.23 | 1.06 | 1.15 | 1.43 | 0.86 | 1.21 | 0.15 |
| Mean | 1.04 | 0.99 | 0.86 | 0.84 | 0.86 | 0.78 | 0.90 | 0.10 |
| s.e. | 0.11 | 0.09 | 0.11 | 0.09 | 0.11 | 0.09 | 0.11 | |

Higher OC is associated with greater nutrient availability, structural stability and water holding capacity (Peverill et al., 1999). Critical values for OC are hard to define, although Kay and Angers (1999) suggest that surface soil OC < 1% may constrain yields to less than the potential based on rainfall. Five of the EIP watersheds had OC ≤ 1% through most of the toposequence.

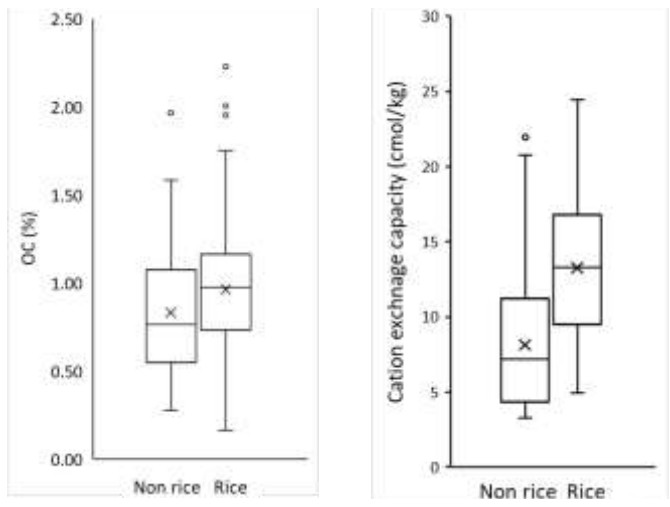


**Fig 3.** (a) Soil organic carbon and (b) cation exchange capacity across seven EIP watersheds.



The OC content of non-arable uplands was close to each watershed mean, despite these being relatively undisturbed, uncultivated areas that we expected to have amongst the highest OC in each watershed. Homestead land also had no greater OC than elsewhere, despite homesteads being the destination for all harvested materials including crop residues. Rice soils had higher mean OC than non-rice soils (0.96 $v$ 0.83%, P<0.01), but there was substantial variation (Fig. 3a). Amongst rice soils, there was a trend towards OC increasing down the toposequence (Table 2).

**3.3 Cation exchange capacity (CEC)**

Mean CEC over all fields was 10.7 $cmol_c/kg$, but it varied among watersheds from lows of 6.5 and 8.2 $cmol_c/kg$ at Khunti and Lohardaga, to a high at Bankura of 16.4 $cmol_c/kg$ (Table 3). The mean CEC of soils used for rice production (13.3 $cmol_c/kg$) was substantially higher (P<0.01) than for non-rice soils (8.1 $cmol_c/kg$) (Fig. 3b), and the CEC of rice soils increased substantially down the toposequence (Table 3).

**Table 3.** Cation exchange capacity ($cmol_c/kg$) of soils in seven EIP watersheds

| Watershed | Rice land | | | Non-rice land | | | Mean | s.e. |
|---|---|---|---|---|---|---|---|---|
| | Lowland | Medium lowland | Medium upland | Arable upland | Non arable upland | Homestead | | |
| Gumla | 13.1 | 9.0 | 8.1 | 9.5 | 9.1 | 10.9 | 10.0 | 0.95 |
| Lohardaga | 10.4 | 12.8 | 8.0 | 6.3 | 7.7 | 3.8 | 8.2 | NS |
| W Singhbhum | 19.1 | 12.7 | 12.0 | 4.2 | 5.8 | 8.8 | 10.4 | 1.38 |
| Khunti | 11.1 | 7.3 | 7.8 | 4.4 | 4.2 | 4.2 | 6.5 | 0.86 |
| Dumka | 18.5 | 14.9 | 12.7 | 5.0 | 7.2 | 4.2 | 10.4 | 1.60 |
| Godda | 16.4 | 15.3 | 14.5 | 10.3 | 11.4 | 9.8 | 13.0 | 1.22 |
| Bankura | 21.8 | 17.5 | 15.1 | 18.7 | 16.7 | 8.6 | 16.4 | 1.69 |
| Mean | 15.8 | 12.8 | 11.2 | 8.3 | 8.9 | 7.2 | 10.7 | 1.08 |
| s.e. | 1.01 | 0.81 | 1.39 | 0.93 | 0.98 | 1.26 | 1.19 | |

CEC is important as it influences nutrient (cation) retention and provides a buffer against soil acidification (Ketterings et al., 2007). To put the EIP values into perspective, most agricultural soils fall in a range between 2 and 40 $cmol_c/kg$, with CEC increasing with clay fraction, organic matter concentration and pH (Peverill et al., 1999).

CEC across the seven watersheds increased significantly with OC ($R^2$ = 0.45, P<0.01) and (weakly) with $pH_w$ ($R^2$ = 0.22, P<0.05) (Fig. 4), explaining the higher CEC of rice-growing land (Fig. 3b).





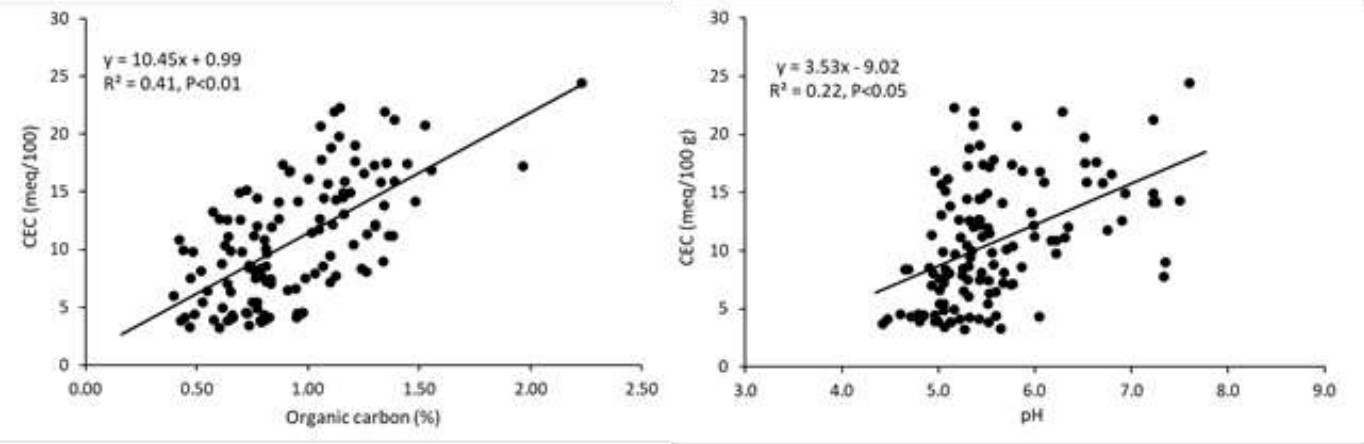

**Fig 4.** The relationship between cation exchange capacity and organic carbon and pH in soils across seven EIP watersheds.

### 3.4 Available phosphorus

The mean concentration of available soil P averaged 4.3 mg kg$^{-1}$ over all watersheds and land classes (Table 4). We expected higher soil P (and OC, CEC) in the absence of cultivation in non-arable uplands, but this was not the case, with mean P of only 3.2 mg kg$^{-1}$. As with other soil fertility indicators, the results for soil P were highly variable, even within land class. Soil P values overall varied from not detectable to 6.4 mg kg$^{-1}$, *excluding* statistical outliers with up to 29 mg kg$^{-1}$ (Fig. 5a).

**Table 4.** Available soil phosphorus (Bray-P, mg kg$^{-1}$) in seven EIP watersheds.

| Watershed | Rice land | | | Non-rice land | | | | |
|---|---|---|---|---|---|---|---|---|
| | Lowland | Medium lowland | Medium upland | Arable upland | Non arable upland | Homestead | Mean | s.e. |
| Gumla | 1.3 | 5.0 | 3.3 | 4.3 | 2.9 | 7.3 | 4.0 | 1.5 |
| Lohardaga | 2.1 | 3.6 | 2.4 | 2.5 | 1.2 | 3.8 | 2.6 | NS |
| W Singhbhum | 0.4 | 4.3 | 12.4 | 6.0 | 5.0 | 7.6 | 5.9 | 2.1 |
| Khunti | 2.1 | 1.4 | 0.6 | 4.8 | 1.9 | 1.0 | 2.0 | NS |
| Dumka | 1.1 | 2.3 | 2.3 | 22.2 | 1.1 | 3.3 | 5.4 | 1.6 |
| Godda | 1.7 | 5.3 | 1.6 | 26.6 | 7.8 | 4.2 | 7.8 | 1.8 |
| Bankura | 1.6 | 2.1 | 5.8 | 2.1 | 2.3 | 1.3 | 2.5 | 0.6 |
| Mean | 1.4 | 3.4 | 4.0 | 9.8 | 3.2 | 4.1 | 4.3 | 1.5 |
| s.e. | 0.4 | 0.9 | 1.7 | 2.1 | 1.1 | 1.2 | 1.4 | |

Concentrations of P were highest in the arable uplands of Dumka and Godda and, less so, the homestead land of Gumla and West Singhbhum and the medium-uplands at West Singhbhum and Bankura. The relatively higher concentrations were found





in small areas where maize (*Zea mays*) or vegetables were grown in the monsoon cropping season. This included two fields in medium-upland in West Singhbhum and Bankura that had recently been converted from rice to vegetables. Maize and vegetables are generally fertilised with di-ammonium phosphate (DAP) or a compound fertiliser containing N, P and K. Available P in rice fields tended to decrease lower in the toposequence (Table 4). In lowlands, the area traditionally used for rice, the P concentration was 1.4 mg kg$^{-1}$ across watersheds. Over all rice fields, 75% had P values <3.6 mg kg$^{-1}$ (Fig. 5a).

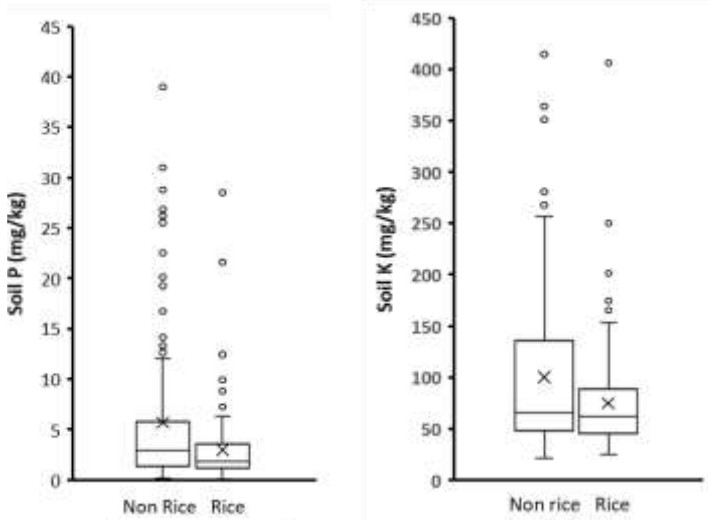


**Fig 5.** (a) Available P and (b) Extractable K concentrations across seven watersheds.

Crops respond to P-fertiliser when Bray-P is less than around 10 mg kg$^{-1}$ for flooded rice and 20 mg kg$^{-1}$ for other crops (Bado et al., 2008; Dodd and Mallarino, 2005; Mallarino et al., 2013; Sahrawat et al., 1997). When <5 mg kg$^{-1}$, transplanted rice yields, without fertiliser, may be less than a third of potential (Bado et al., 2008). The values for Bray-P in Table 4 suggest that

all crops grown anywhere in the landscape (except arable uplands at Godda and Dumka) will be very highly responsive to P-fertiliser, even with the lower critical value for rice under flooded conditions (Seng et al., 2007). In short, these soils are almost universally acutely deficient in P. This is supported by limited on-farm research in Purulia District of the eastern EIP showing large responses to P-fertiliser in rice (Cornish et al., 2010), and even larger responses in mustard (*Brassica juncea*) and wheat (*Triticum aestivum*) following rice, where yields without P fertiliser were <10% of well-fertilised fields (Cornish et al., 2015b).

**3.5 Extractable soil potassium**

The mean extractable K concentration was 88 mg kg$^{-1}$. This varied among watersheds, from 50 mg kg$^{-1}$ to 164 mg kg$^{-1}$, and among land classes, from 64 mg kg$^{-1}$ in lowland to 128 mg kg$^{-1}$ in arable upland (Table 5). Concentrations were highest in arable uplands in Khunti, Dumka and Godda, homestead land at Gumla, and medium-upland at West Singhbhum (skewed by one high field). This pattern mirrored available soil P and, as previously noted, may reflect the use of fertilisers on maize and

vegetables. Non-rice fields overall had more K than rice fields (100 *v* 75 mg kg$^{-1}$, P<0.01) (Fig 5b). The K in rice fields





generally decreased lower in the toposequence. The outliers in the non-rice fields in Fig. 5b were almost all associated with arable uplands at Godda, where high K seems to be a feature of this land class for unknown reasons (Table 5).

**Table 5.** Extractable potassium (mg kg[-1]) in seven EIP watersheds.

| Watershed | Rice land | | | Non-rice land | | | Mean | s.e. |
|---|---|---|---|---|---|---|---|---|
| | Lowland | Medium lowland | Medium upland | Arable upland | Non arable upland | Homestead | | |
| Gumla | 92 | 131 | 143 | 197 | 179 | 240 | 164 | 23 |
| Lohardaga | 58 | 86 | 79 | 49 | 66 | 53 | 65 | NS |
| W Singhbhum | 80 | 66 | 161 | 60 | 60 | 83 | 85 | 26 |
| Khunti | 51 | 45 | 53 | 65 | 43 | 45 | 50 | 6 |
| Dumka | 41 | 41 | 76 | 194 | 80 | 63 | 83 | 15 |
| Godda | 71 | 74 | 37 | 265 | 88 | 56 | 99 | 22 |
| Bankura | 56 | 60 | 74 | 69 | 106 | 50 | 69 | 13 |
| Mean | 64 | 72 | 89 | 128 | 89 | 84 | 88 | 16 |
| s.e. | 6 | 11 | 26 | 21 | 11 | 16 | 18 | |

The data show large variation within and between watersheds, with half of all fields having less than ~60 mg K kg[-1]. The significance of these values depends on the critical values used for interpretation. These depend on the extractant, soil type, sampling depth, crop species and crop demand for K. Peverill et al. (2005) suggest categories of low (<80 mg kg[-1]), medium (80-200 mg kg[-1]) and high (>200 mg kg[-1]), but highlight the poor predictive power of soil K tests. A review of Australian field experiments suggests a critical value closer to 50-60 mg kg[-1] for moderately acid soils, based on bicarbonate-extractable K (Brennan and Bell, 2013). On this basis, extractable K would be low to marginal except for Gumla, the medium-upland in West Singhbhum (skewed by one outlier) and arable uplands in Godda and Dumka.

P and K tended to vary together, but only weakly ($R^2 = 0.34$, Fig. 6).

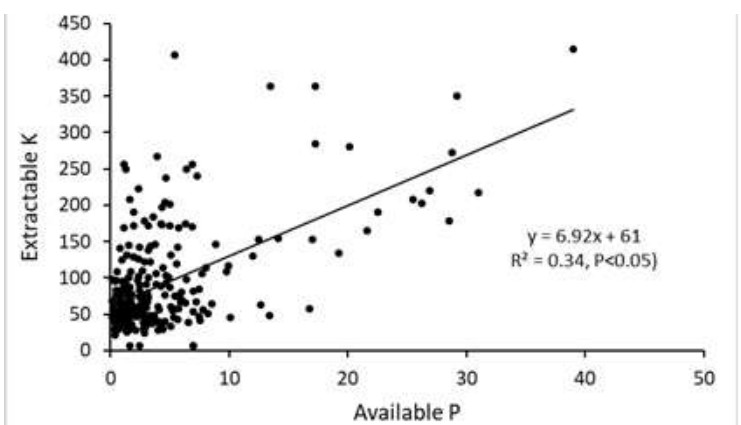

$y = 6.92x + 61$
$R^2 = 0.34$, $P<0.05$)

**Fig. 6**. The relationship between available P (mg kg[-1]) and extractable K (mg kg[-1]) in soils across seven EIP watersheds.





## 4. General discussion

Prior to this study, published information on soil fertility on the EIP was restricted almost entirely to large-scale soil maps, leading to generalisations about acid, infertile soils (Edmonds et al., 2006; Sikka et al., 2009), and in turn to generalised fertiliser prescriptions. These generalisations fail to capture variation down the toposequence within watersheds and between individual fields. Our findings for uncultivated uplands confirm that the soils are inherently acid and infertile, but also reveal systematic variation associated with land use and location in the toposequence.

**4.1 Differences along the toposequence**

Soils from the three classes of upland, where rice is not grown, had significantly lower pH, CEC and OC, but more P and K, than the three classes of rice-growing fields lower in the toposequence. Amongst rice fields, those lower in the toposequence tended to be less acid and have higher OC and CEC, but less P and K. The trends were generally consistent amongst the watersheds and consistent with the findings of Cornish et al. (2010) for two watersheds in Purulia District.

The low soil P and K in lowlands suggests that the higher rice yields reported by farmers for this land class reflect more reliable water rather than greater soil fertility. Nevertheless, farmers regard lowlands as their 'most fertile' land.

Our findings contrast with those of Fuwa et al. (2007), who reported fertility *increasing* down the toposequence in a single village. No details for sampling and analysis were provided. In another study, in four villages each in Thailand and Vietnam, Boling et al. (2008) reported higher OC and pH in rice fields lower in the landscape as in our study, no trend in soil P, and an

increase in soil K down the toposequence but only in Indonesia.

Differing toposequence trends could reflect numerous variables, including underlying geology and native soil fertility, cutting and filling to terrace land for rice, the length of time land has been used for rice, and the level of fertiliser inputs over that period. Amongst the EIP watersheds reported here, only the underlying geology will vary greatly, possibly explaining the relative consistency of our findings. The watersheds in our study are characterised by their low inherent fertility (evident in

the uncropped uplands), low nutrient inputs, and long histories of rice in lowlands and medium-lowlands but not in the medium-uplands that are still being developed by marginal farmers to increase their area of lowland (transplanted) rice. Our findings will likely apply to any undulating landscape in Asia that produces rice and has the characteristics just described.

One of our aims was to evaluate the possibility of basing fertiliser recommendations on topographic position, as suggested by Boling et al. (2008). Our data show this is not feasible because of the large variability between fields within land

class/topographic position, implying the need to avoid broad prescriptions and to manage nutrients and fields individually. However, field-specific management raises the question of how farmers can vary inputs to meet crop requirements in the absence of soil testing, which they cannot afford even if it is available and reliable. The answer may lie in simple farmer 'test-strips' (Cornish et al., 2015b) or the 'omission trial' recommended by IRRI (2019b). These can be used effectively by farmers given some technical support and a participatory learning process (Pretty, 1995).





## 4.2. Nutrient flows in the landscape and effects on soil fertility and rice yields


The vast majority of smallholders on the EIP are subsistence farmers, so almost all food is produced for home consumption and little is exported from the farm (and watershed). Nutrients removed from cropped land and grazed/forested areas are relocated to homestead areas where the grain is consumed, straw is utilised for roof thatch, animal bedding and feed, and the manure collected from grazed land is used for fertiliser or for fuel (along with firewood). Organic materials and ash collected

from around the homestead are then partially returned to cropped areas as manure, compost and ash. These nutrient flows are shown in Fig. 7. Upland soil erosion is a significant potential source of nutrients for crops lower in the landscape, and animal carcases are a significant sink. Nutrient import in subsidised food may be significant for the poorest families.

Whilst these systems appear to be relatively 'closed' compared with market-oriented agriculture, nutrients are still lost in runoff and in the small amounts of produce sold (sometimes including old cows and other animals). Nutrients may also be

imported in fertiliser, but these inputs are relatively small compared with elsewhere in India (FAO, 2005; Anon., 2013), with rates estimated from these reports to be 40 kg N ha$^{-1}$, 5 kg P ha$^{-1}$ and 8 kg K ha$^{-1}$ per rice crop.

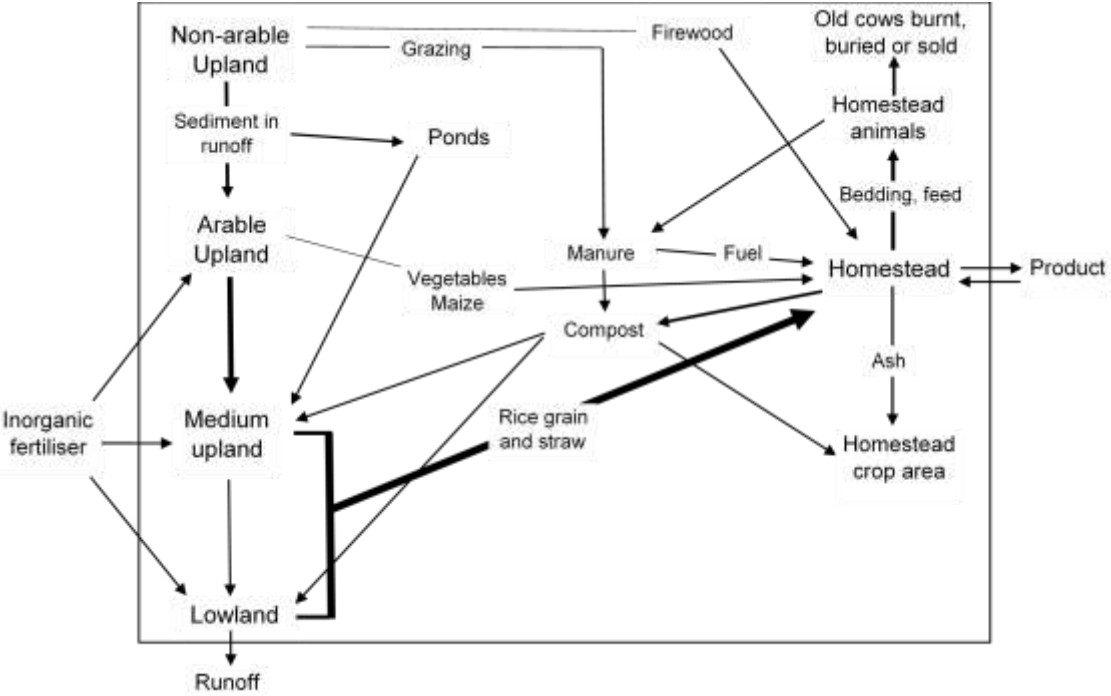

**Fig. 7**. Indicative nutrient flows. Line weight reflects relative magnitude (see text for details).

Given the homestead is the focus of nutrient flows we expected to find OC, CEC, P and K to be highest in this land class.

However, P and K tended to be highest in arable uplands and some homestead land, whilst OC and CEC were highest in rice land. Our tentative explanation is that farmers prioritise rice for their compost (in addition to inorganic fertiliser), as it is the



foundation for food security. Other crops receive mainly inorganic fertilisers as finances allow. The use of compost on rice land will result in relatively higher OC and CEC (Figs. 3, 4), whilst using inorganic fertilisers and ash on non-rice crops such as maize and vegetables will help to maintain relatively higher soil P (Fig. 5a) and K (Fig. 5b) where these crops are grown,

i.e. around the homestead (e.g. Gumla, Lohardaga), arable uplands (Dumka, Godda), and selected medium upland fields converted from rice to vegetables (West Singhbhum, Bankura).

That arable uplands had more P and often K than non-arable uplands (except Bankura) suggests that rates of inorganic fertiliser were sufficient to maintain the nutrient balance, albeit at low concentrations of P and K with low yields. In rice, it appears that recycled nutrients in compost plus the small inputs of inorganic fertiliser are not sufficient to balance removal, and further

declines in soil nutrient content might be expected with current practices. Further research on field and watershed-scale nutrient balances are required to evaluate these propositions.

Low fertility resulting from inadequate fertiliser appears to be the primary cause of low rice yields in the region (<1.9 t ha$^{-1}$) (IRRI, 2019a), rather than insufficient water. There is reliable water for rice in lowlands and in all areas for crops that do not depend on flooding (Cornish et al., 2015a, b).

We now consider which of the major nutrients (N, P, K) is the main constraint to rice yield. Flooded rice has historically been grown as a subsistence crop on alluvial or colluvial lowlands with no external inputs. Yields were sustained with N from free-living N-fixation (Watanabe and Roger, 1984) and P from sediment in runoff from uplands. Although the amount of N fixed is difficult to quantify, Watanabe and Roger (1984) suggest it may be around 80 kg ha$^{-1}$ per crop. If N were the primary nutritional constraint to yield, and rice requires 15-20 kg N t$^{-1}$ (IRRI 2019b), then yields of 4 t ha$^{-1}$ should be sustained even

without N-fertiliser. As farmers in the region apply on average 40 kg N ha$^{-1}$ there should be ample N for 4 t ha$^{-1}$ yields, yet they average less than half this.

From our soil analyses (Table 4), we hypothesise that P is the primary nutritional constraint to rice yields in almost all fields unless P-fertiliser is supplied at high rates. Current rates appear to average only around 5 kg P ha$^{-1}$ per crop for subsistence farmers (FAO, 2005; Anon., 2013) which, assuming 100% fertiliser efficiency, is enough for a crop of only 2 t ha$^{-1}$ (IRRI,

2019b), i.e. the present regional rice yield.

We hypothesise that P-fertiliser is required now, when it was not required in the past, because progressive terracing of hillslopes has intercepted runoff from uplands and, with sediment deposition, deprived lowlands of their traditional source of P. With reduced sediment-P input, and with low rates of fertiliser-P, a rundown of soil-P is inevitable and unsustainable. This will be most evident in lowlands as the oldest cropped land and the least likely to received sediment-bound P. We suggest that

substantially increased P-fertiliser inputs are needed to efficiently use the available water and the N described above.

K appears worthy of further attention, even if it is not currently the primary nutritional constraint. K concentrations appear to be low to marginal, and present practices return insufficient K to rice land to maintain soil K. Rice removes ~5 kg K t$^{-1}$ grain unless straw is retained (IRRI, 2019b), suggesting a fertiliser replacement value of 20 kg K per crop if yields are 4 t ha$^{-1}$. Farmers on average apply 8 kg K per crop so rundown of soil K seems inevitable. The greatest risk of K deficiency is in the

lowlands that have been cropped for the longest period of time and have the lowest concentrations of extractable K (Table 5).





In the absence of reliable dose-response data and critical soil K values, there would be merit in farmers using test strips to assess responses to K-fertiliser, giving priority to lowlands. Or they could simply commence K-replacement on lowland.

It is clear that higher yields and sustainable crop production will only be achieved with substantially increased fertiliser use. This will present a challenge to risk-averse subsistence farmers, who also have limited access to credit for inputs. The field
test strips and omission trials referred to earlier could assist farmers to assess nutrient responses on their own land and gain confidence whilst taking manageable risks.

### 4.3 Soil acidity

Results confirm the widely held belief that EIP soils are acid (e.g. Edmonds et al., 2006; Sikka et al., 2009), although there is significant variation amongst the watersheds with some near neutral (Godda) and others requiring remediation (e.g. Khunti).
Uniform recommendations to lime EIP soils would be inappropriate because only 14% of all fields had $pH_w$ <5.0. Although lime would not normally be considered for lowland rice at this pH, any non-rice crops following rice may require lime.

A further 44% of soils are moderately to slightly acid ($pH_w$ 5.0-5.4), and for these fields there is a need to at least minimise further soil acidification. A major cause of soil acidification is the use of ammonium-based fertiliser (Helyar and Porter, 1987), such as diammonium phosphate (DAP) which is widely used as a pre-plant fertiliser for all types of crops, albeit as low rates
currently. Our results suggest that inputs of N should be restricted to urea, which is less acidifying. P must then be supplied as superphosphate which is essentially non-acidifying (Helyar and Porter, 1987). We also recommend that K, if required, be applied as muriate of potash or potassium sulphate, allowing farmers to vary individual nutrients according to requirements and minimise soil acidification. The use of superphosphate and potassium sulphate will also help to address any concern about S, which is not addressed here.
Fortunately, $pH_w$ increased with depth at the one watershed where this was studied, and it is unlikely that subsoil amelioration will be needed (Helyar and Porter, 1987).

### 5. Conclusions and Recommendations

Fields without rice, higher in the toposequence, had lower $pH_w$, OC and CEC but higher P and K than rice fields. Amongst rice fields, those lower in the toposequence were less acid and had higher OC and CEC, but less P and K. These toposequence
differences may apply to other undulating landscapes in Asia exhibiting the characteristics of our present study, i.e. low inherent fertility, low nutrient inputs, and a long history of rice in lowlands and medium-lowlands, but not in medium-uplands that are still being developed as marginal farmers increase their area of lowland rice.

Despite these toposequence differences, fertility indicators varied so much within areas defined by land class and hydrology that toposequence position cannot be used to better target fertiliser recommendations.
For most arable fields and for all crop types, yield improvement requires the application of much higher rates of P and possibly K fertiliser (N is deficient but was not studied). We hypothesise that for many years rice was grown successfully on inherently

infertile lowland soils, without inorganic fertiliser, because eroded sediment from uplands annually renewed the soil P (and other nutrients), a source now diminished by terracing of uplands. This has led to declining soil P and the need to greatly increase fertiliser rates. Soil K is not so critically deficient, but it too is declining and K-replacement needs to be considered.

The present cropping system is clearly unsustainable and unable to efficiently use the available water or even the N supplied by free-living N-fixation.

Soils are mostly acid, but only 14% of fields ($pH_w<5.0$) warrant liming. A further 44% are at risk from further acidification, so we recommend replacing fertilisers containing $NH_4^+$, such as DAP, with less-acidifying sources such as urea (for N) and superphosphate (for P, S), enabling farmers to target fields with the N, P and K required, with minimal effect on soil pH.

Variation within fertility indicators and weak associations between them show that nutrient-use needs to be field-specific. We suggest introducing farmers to 'strip tests' or 'omission trials', allowing them to make their own field-by-field assessment of nutrient and lime requirements. Engaging risk-averse farmers in a participatory learning process should help them to take the difficult and expensive decision to substantially increase fertiliser rates.

**Data Availability**

Data are freely available for download from Harvard Dataverse. https://doi.org/10.7910/DVN/UUEF6J

**Author contributions**

PC was overall project leader. All authors contributed to experimental design. AK led the development of site selection criteria, site selection, and discussions with farmers. SD oversaw all sample collection, processing and despatch to the laboratory. PC prepared the manuscript with comments by AK and SD.

**Acknowledgements**

This research was funded by the Australian Centre for International Agricultural Research (Project LWR/2002/100). Dr. K. S. Reddy, Indian Institute for Soil Science, arranged for soil analyses. Dr S. Kumar, Indian Council for Agricultural Research (Ranchi), arranged assistance with field sampling, Messrs Mahato and Lakhi (PRADAN) provided invaluable assistance with sample processing prior to analysis and Prof Richard Bell (Murdoch University) commented on the draft manuscript. Many
farmers participated in the research, and without access to their land, their ideas, and their assistance, the work would not have been possible.

**Competing Interests**

The authors declare that they have no conflict of interest



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
