# Peer review of "Soil fertility along toposequences of the East India Plateau and implications for productivity, fertiliser use and sustainability"

_SOIL, 2019_

## Short Comment (SC1) · 9 Feb 2020

This paper very well explains about the variability among the land class and within land class. It is important to know for the farmers and extension workers who need this information for nutrient application in field. Most of the time the recommendations are crop specific and not the land class specific. So this paper puts emphasis on to consider the nutrient status variability in the land class while suggesting nutrient application for a particular crop.

---

## Referee Comment (RC1) · Anonymous Referee #1 · 15 Mar 2020

GENERAL COMMENTS

In this work authors address a crucial issue for the agricultural production and in this specific case for the rice cultivation in different areas of East India Plateau: the soil fertility along the toposequence. Indeed, the manuscript develops on the hypothesis that toposequence position affects soil fertility and aim at providing basic information for future research on the management of soil fertility and plant nutrition. Authors also attempt to provide technical recommendations on fertilizer use. Accordingly, this contribution falls within the scope of Soil Journal.

While acknowledging the effort put by authors in developing this work, which can con-

tribute to better understand the dynamics underlying the creation and the maintenance of soil fertility, some shortcomings have been detected in the manuscript. First, the title is promising to analyze the soil fertility implications in terms of sustainability. Nevertheless, this very important concept is never addressed in the paper and it is not clear in which terms sustainability is taken into consideration in this work. Besides, the rice productivity, another concept included in the title, is slightly addressed and is indeed limited to only few yield data that are mainly mentioned in the discussion. Yet, a description of the rationale backing the selection of the analysed soil parameters is missing. To this regard would be necessary to briefly explain the influence/contribution of each soil parameters to soil fertility. Such rationale would allow readers to better understand the discussion and the recommendations about soil fertility raised by the authors. Finally, this work mainly aims at providing a picture of the variability of the soil parameters between the different land classes. Accordingly, the figures depicting the final results should clearly show such soil parameters variability when instead now is limited to differences between Rice areas and No Rice areas. This work has been well written in a very good English. Nevertheless, in few circumstances the manuscript is not very clear. However, this paper needs to be improved with a major revision before being published in this Journal. Following, specific comments are provided hoping that these can be of help to improve the manuscript.

SPECIFIC COMMENTS

From Line 98 to Line 101: The definition of the land classes is a bit confusing. Are the three categories of upland land-classes? In case, please specify.

Line 99: By definition Homestead includes land that is cropped. Should (i) be "land belonging to the homestead...", rather than "land near to the homestead..."?

Line 99: Please, could clarify what the sentence "to which harvested materials are generally taken" means?

Line 103: Can authors be more specific briefly explaining why "Non-degraded nonarable uplands provide an indication of inherent soil fertility"?

Line 106: To make reader to understand the representativeness of the soil samples, the average size of field and the main characteristics of transect (e.g. length and/or distance between the four locations) applied for sampling should be mentioned.

Line 111: It is not clear why the authors refer to unpublished data that furthermore refer to a site that is not under analysis in this work. If this set of data should substantially contribute to the understanding of this work, then should be clearly stated and explained in the text.

Line 120: "A small number of our samples. . ." Could be more specific by indicating the total number of samples and what small number means?

From line 125: It seems that from this line to Line 133, the text is devoted to explain the "statistical analysis". It is suggested to include this part in an independent paragraph or sub-paragraph (2.3.1) of soil analysis.

Line 140: The minimum values of the pH depicted in Figure 2a for non-rice and rice fields seems to be different when in the text the two type of soils have the same value, i.e. 4.4.

Lines 148, 152, 155: the Purulia District is often mentioned in this work notwithstanding it is not amongst the District under analysis. Authors should clarify the role of this District in this work (comment linked to Line 111).

Lines 161 and 162: Because of the relevance of this topic especially in this work, it is suggested to provide more recent literature in addition to Peverill et al. 1999 and Kay and Angers 1999.

Line 164: It would be interesting to know the value of "potential yield based on rainfall". This would contribute to better understand the discussion about rice productivity.

Line 169: Can authors specify why homestead land "being the destination for all harvested materials including crop residues" was expected to have a higher OC content?

Line 172:§3.3 Cation exchange capacity. Although the Fig. 4 provides readers with very interesting correlations between CEC and OC and pH, it is suggested, in addition, to display box plots as already done for OC and pH (further following the suggestion "Lines 155 (Fig. 2), Line 165 (Fig. 3), etc." reported in section "Figures" of this review).

Line 182: Please, more recent reference(s) in addition to Peveril et al., 1999, is/are desirable.

Line 204: ". . .may be less than a third of potential". Also in this case it would be useful to have some data (please, see comment Line 164).

Line 210: §3.5 Extractable soil potassium. Please, refer to the suggestion raised for Line 172.

From Line 236 to Line 245: In this part of the text, authors state that in contrast to other authors (Fuwa et al 2007), the fertility decreases down the toposequence and that the higher fertility of lowlands claimed by farmers is likely due to a higher water availability that overweighs the very low P and K content. Nevertheless, the lowlands have registered also an higher OC and CEC that are key drivers for determining soil fertility just as confirmed by this work in which these elements were carefully taken into consideration. To avoid confusion in the interpretation of the results, authors must provide (possibly in the Introduction) a clear definition of what soil fertility is in this work and to which extent the different elements (OC, CEC, K, etc.) taken into consideration contribute to its setting up.

Line 248: Since the "underlying geology" seems to play a crucial role in this work, some key detail about the geology of the analysed EIP watersheds should be provided to readers.

From Line 251 to Line 252: "Our findings will likely apply. . .", then what raised for Line 248 become very important.

From Line 257 to Line 259: These are relevant recommendations (i.e. the 'test strips', the 'omission trial'). Nevertheless, they are only mentioned and referenced but an interesting discussion is missing. Authors should spend some word to briefly explain their recommendations. Some detail about the learning process is also desirable.

Line 262: Although the meaning of Nutrients is well known, probably it would be more clear to specify which nutrients are taken into consideration in this discussion.

Line 266: "...and animal carcases are a significant sink". Please, can clarify? Besides, animal carcases are not included in Fig.7 (the category "Old cows burnt, buried or sold" does not contribute to the flows).

Line 270: imported in fertilizer or imported with fertilizer?

Line 275: ...farmers prioritise rice for their compost... This is an important statement and would be desirable to support it with some reference.

Line 283: Some data about low yields and rice yield in general in the EIP should be provided.

Line 283: Should "In rice..." be "In rice land.."?

Line 294: "...rice requires 15-20 kg N t-1" , please specify what tons refer to (grain?).

From Line 293 to Line 310: The assumptions and related calculations made in these lines seem to be approximate. From line 293, authors refer to a direct correlation between the quantity of N fixed, the average quantity required by the crop and its yield. Nevertheless, the various "destinies" of the Nitrogen in soil (i.e. leaching, denitrification, immobilization, etc.) seem to be neglected. Same consideration applies for P and K. This part should be carefully revised.

Line 313: "...sustainable crop production..." Sustainable in which terms?

Line 315: As previously commented (Lines 257-259), test strips and omission trials should be briefly described.

To make effective the interesting discussion that authors raised from line 318 to line 331, it is suggested to include a paragraph/table (maybe in material methods) in which the main fertilization techniques (type of fertilizer, quantities and timing) currently adopted by farmers are summarized. This would allow readers to better understand the discussion and the authors' recommendations.

From Line 323 to Line 325: A major cause of soil acidification is the use of ammonium-based fertilizer... Please, make clear if this is a general statement or specifically refer to the soils under analysis. For instance, it is not clear if DAP is used in the fields under study and if the related applied quantities can actually determine the acidity of such soils.

Line 345: "The present cropping system is clearly unsustainable..." In which terms is unsustainable?

- REFERENCES

Please, format the references according to the Journal requirements.

Please, verify "Kay and Angers", 1999 in the text, 2000 in References.

- FIGURES

Captions should be self-explanatory avoiding to send the reader to the text (e.g. Fig. 7).

The quality of the Figures layout should be improved. It is currently low.

A figure/map depicting the sites would be appreciated.

Figure 1. Whether possible, it would be helpful to include also the homestead in this figure, making clear its position along the toposequence.

Line 42: Figure 1. Should (after Cornish et al., 2015a) be (Modified from Cornish et al., 2015b)?

Lines 155 (Fig. 2), Line 165 (Fig. 3), Line 201 (Fig. 5): In order to appreciate the influence of the toposequence to soil fertility, authors identified six land classes. To this regard, it would be effective to show the variability of the analysed soil characteristics/properties (pH, OC, etc.) between the six land classes (e.g. one figure depicting one box plot for each land class) in addition to that between the No-rice and Rice areas (Figures 2a, 3a, 3b, 5a and 5b).

Figure 4 and Figure 6: Please, specify what the depicted correlation points refer to (are they the samples collected in the 252 fields?).

- TABLES

Brief but explanatory captions should be included.

For the sake of a better understanding, Tables from 1 to 5 should be set with the same schematic logic displayed in Figure 1 where the toposequence goes from the higher to the lower level moving from left to right.

---

## Referee Comment (RC2) · Richard Bell (Referee) · 18 Mar 2020

I'm satisfied that the paper makes a useful contribution to the literature by drawing attention to new understanding about landscapes that are quite common across Eastern India and SE Asia. In these regions, patterns of hydrology and crop/ variety choice are related to land type in relation to a toposequence. It is assumed that soil nutrient status would also follow this pattern. Cornish et al. have found contractors evidence which can trigger a re-think about nutrient management in these landscapes.

On the attached file I have added comments and suggestions for revised wording, to improve clarity.

[Figure]

Please also note the supplement to this comment:
https://www.soil-discuss.net/soil-2019-92/soil-2019-92-RC2-supplement.pdf

**Supplement:**

[revised manuscript text omitted]

---

## Author Comment (AC1) · 25 Mar 2020

Thank you for your comment. When soils are as variable as we found here, the challenge is to move away from prescriptive recommendations based on a belief that the soils are infertile, to an approach that attempts to meet the actual requirements of individual fields/crops. In the ideal world, we might use soil tests to tailor a fertiliser mix to each crop type and field, but in Eastern India (and elsewhere in S and SE Asia) this is not practical because reliable and timely soil testing services are unavailable and/or unaffordable, and in in any case, the results can be hard to interpret. We believe that, in the right learning environment, farmers can use simple fertiliser test strips or omis-

sion trials to effectively explore fertiliser requirements. Whatever approach is taken to extension the message is clear, that most farmers at present achieve yields that are well below the rainfall-determined potential because of nutrient deficiencies, and the cropping systems are unsustainable because of ongoing nutrient depletion.

---

## Author Comment (AC2) · 3 Apr 2020

We thank the referee for their extensive, thoughtful comments that will lead to an improved manuscript. We will address the general and substantive specific comments here, and deal with other specific comments when we revise the manuscript.

The referee notes that the title refers to sustainability, but then says "this very important concept is never addressed in the paper and it is not clear in which terms sustainability is taken into consideration in this work". In response, we note that the Abstract reports the low and evidently declining concentrations of soil P and K, and concludes that "Fertiliser-use must increase substantially to sustain the system". This argument

regarding the sustainability of the cropping system is fully developed in lines 260-316 of the Discussion. In the Conclusions we say the present cropping system is 'unsustainable' because the soil fertility is mostly low and continuing to decline because fertiliser rates are too low. We need to specify in our revision that here we are referring to P and K.

The referee also notes reference to "productivity" in the title and goes on to say that "rice productivity . . . is limited to only few yield data that are mainly mentioned in the discussion". Yields were not actually measured in the study and this is why they are not referred to in the results; however, the Introduction (lines 48-57) makes it clear that the context for the work is the general low rice productivity on the Plateau that is related to soil fertility rather than a lack of rainfall. This is low "water productivity", a theme that is taken up at line 287-289 and again at line 345 where we conclude that "The present cropping system is clearly unsustainable and unable to efficiently use the available water".

The referee notes that ". . . a description of the rationale backing the selection of the analysed soil parameters is missing". The rationale is given in the Methods that state "Earlier research in Purulia District, West Bengal (Cornish et al., 2010) led to a focus on soil pH, organic carbon (OC), cation exchange capacity (CEC) and the macronutrients phosphorus (P) and potassium (K) as indicators of chemical fertility". We will add into the Discussion that the 1:50,000 soil maps we refer suggest that boron might be widely deficient (and variable), but we did not assess this.

The referee suggests that "the figures depicting the final results should clearly show such soil parameters variability when instead now is limited to differences between Rice areas and Non-Rice areas". We note that all of the figures in the paper are box and whisker plots that provide detail on the variability within the aggregated land use categories of rice and non-rice, whilst the tables all give standard errors for each analyte that apply to land class, watershed, and the data overall.

The referee refers to areas of text that are unclear – these will be attended to when we revise the paper. Referee 2 also cited instances where the text could be clearer. Regarding specific comments on the paper, we plan to respond positively to all of the suggestions when we revise the paper, but offer the following comments on a few of the more substantive matters.

Line 103. Non-degraded uplands provide an indication of inherent fertility because they have not been subjected to nutrient removal through cropping or extensive soil erosion that is evident on degraded uplands. They are certainly not a perfect indicator of inherent soil fertility, but under the circumstances they provide a useful indication.

Line 106. As a matter of general interest for readers, the fields referred to are generally very small, sometimes less than 200 m2 in the case of rice on bunded hillslopes, and infrequently more than 1,000 m2. Families own or lease multiple fields, the number, location and quality of which depends on family wealth.

Line 111. Soil acidity is not confined to surface soils, but our resources did not allow for subsoil sampling in the present study. However, extensive soil pH measurements through the profile made in the smaller, preceding study (Cornish et al., 2010) had not been published, so they were included here with a description of the methods. In the absence of complete data, this restricted dataset allowed us to suggest a tentative but important conclusion that remediation of soil acidity may concern only surface soil (line 153). We will expand on this in the revised text.

Line 140. The referee raises a good point. In the text, values were rounded to one decimal point (both 4.4), but the means were actually 4.35 (non-rice) and 4.44 (rice). The 0.9 difference evident in figure 2a is the actual difference. We will amend the text to 2 decimal places.

Line 164. Interesting point. In our previous work we measured yields in hundreds of farm fields and in experiments, with around 7 t/ha as the maximum in both, but rarely in farm fields. Also, in our previous research, we estimated that annual ET of medium

duration rice (125 days) in this region is around 460 mm. If we assume transpiration is around two-thirds of ET, then T is around 280 mm. If transpiration efficiency is 25 kg grain/ha/mm then a potential yield would be around 6.9 t/ha for this class of land and crop duration. We will mention this in Discussion.

Line 172. WE think the referee is asking for a box plot for CEC. This is already given in Fig 3c.

Line 204. The referee requests data, but we did not study crop responses to added P. We discuss the significance of observed P values by referring to the literature (4 references cited).

Lines 236-245. The referee is right that that soil fertility may include a large number of variables, of which OC, CEC, pH, P, K are some – these being selected in our study either because they are universally accepted as being important (OC, CEC) or shown in previous research to be particularly important in the region (pH, P, K). We will clarify this in the Introduction and return to this point in Discussion at lines 236-245.

Line 248. No specific observations were made of site geology. Rather, watersheds representing diverse geologies were selected, based on the descriptions found in the cited reference 'Soil maps, Department of Agriculture, Animal Husbandry and Cooperative. http://agri.jharkhand.gov.in/default.asp?ulink=resources/soilmap.asp. We will explain this in the Methods.

Line 251-252. The point we are making here is that our results were quite consistent across sites, unlike in other studies, and we attribute this to the relative homogeneity of watersheds notwithstanding differences in geology. That is, although geologies varied, the fertility trends generally did not. We will clarify this in the text.

Lines 257-259. Good point. We will elaborate on this in the revised paper.

Line 266. Animals carcases are a nutrient 'sink', where nutrients accumulate and are effectively removed from nutrient cycles. The fate of these nutrients depends on

whether they are sold (exported from watershed) or die and are disposed of within the watershed, but in most cases the nutrients are lost from the nutrient cycle, at least in timeframes relevant to farm management. We will expand on this in Discussion.

Lines 293-210. Noted and will be revised

Lines 318-331. We will comment on this briefly at around line 56 in the Introduction and return to fertiliser types in the Discussion, where appropriate, noting where DAP is most commonly used (DAP being associated with soil acidification).

───────────────────────────────

---

## Author Comment (AC3) · 3 Apr 2020

Referee 2

We thank the referee for the supportive general comments, and the changes suggested in the supplementary comments that will improve clarity and accuracy.

Title change. Agreed

Line 9. The referee suggests 'wetland', and we concur.

Line 16. We prefer to keep the current wording. Management embraces a wider set of options than is implied by the word 'ameliorate', that typically means liming. Manage-

ment may include, for example, the choice of N source and the timing of N application to minimise leaching. We used 'further' intentionally, because acidification has already occurred.

Line 146. We will amend this text.

Line 183 Agreed

Line 226. The relationship is certainly weak as we said, but it is significant ($P<0.05$). It is the weakness that is actually important. We will add here that this was the only significant (but weak) relationship between any of the fertility indicators. We conclude the paper (line 350) by saying that weak associations between the fertility indicators highlights the need for field-specific fertiliser regimens rather than broad prescriptions.

Line 251 Agreed

Line 254. The referee appears to refer to the following paper, that we will now cite: Homma K, Horie T, Shiraiwa T, Supapoj N, Matsumoto N, Kabaki N (2003) Toposequential variation in soil fertility and rice productivity of rainfed lowland paddy fields in mini-watershed (Nong) in northern Thailand. Plant Production Science 6, 147–153. These authors reported, amongst other things, higher OC in rice fields lower in the landscape, as we report.

Line 284. Agreed

Line 299. We agree that fertiliser efficiency will be very much less than 100

Line 329. We will add that B deficiency is widespread according to Anon (2013).

Line 346. Agreed

Line 349. Agreed. We will expand the argument.

---

## Author Response (AR1)

| Referee 1 | | |
|---|---|---|
| Line no. | Comment | Response (line numbers refer to revised MS) |
| General | Referee 1 notes the title refers to sustainability, but then says "this very important concept is never addressed in the paper and it is not clear in which terms sustainability is taken into consideration in this work". | Minor changes made. The original Abstract reports the low and evidently declining concentrations of soil P and K, and concludes that "Fertiliser-use must increase substantially to *sustain* the system". The *revised* Abstract (line 23) and Conclusions (line 349) now specify that we are referring in particular to the unsustainable management of P and K. The argument regarding sustainability of the cropping system is fully developed in terms of nutrient flows and balances in lines 262-320 of the Discussion. The Conclusions says the present cropping system is *'unsustainable'* because soil fertility is low and it continues to decline because fertiliser rates are too low. |
| | Referee 1 notes reference to "productivity" in the title and goes on to say that "rice productivity ... is limited to only few yield data that are mainly mentioned in the discussion". | Minor changes made. Yields were not measured in the study, which is why they are not referred to in the results. However, the Introduction (lines 48-57) makes it clear that the context for the work is the general low rice productivity on the EIP that is related to soil fertility rather than a lack of rainfall. This is low "water productivity", a theme taken up at line 289-291, and again at line 349 where we conclude that "The present cropping system is … unable to efficiently use the available water". |
| | Referee 1 notes that "... a description of the rationale backing the selection of the analysed soil parameters is missing". | Minor change made. The rationale is given in the Introduction, lines 66-68, stating "Earlier research in Purulia District, West Bengal (Cornish et al., 2010) led to a focus on soil pH, organic carbon (OC), cation exchange capacity (CEC) and the macronutrients phosphorus (P) and potassium (K) as indicators of chemical fertility". We have added to the Discussion (lines 319-20) that soil surveys suggest boron might sometimes be deficient, but we did not assess this. |
| | Referee 1 says "figures depicting the final results should clearly show soil parameter variability, when instead now (it) is limited to differences between Rice areas and Non-Rice areas". | No change made. All figures in the paper are box and whisker plots that provide detail on the variability within the aggregated land use categories of rice and non-rice, whilst the tables all give standard errors for each analyte that apply to land class, watershed, and the data overall. |
| | This is a non-specific reference to unclear text | Referee 2 also cited instances where the text could be clearer. These have been attended to in the revised manuscript. |
| | | |
| 98-101 | Definition of land classes | The definition, now slightly expanded, runs to 6 lines (98-104), and references Fig.1 for further information. We feel this is sufficiently explanatory. Ref 2 did not comment. |
| 99 | Homestead land clarification | Not changed. Proximity to the homestead is what matters, not the ownership. |
| 103 | Explain why "Non-degraded arable uplands provide an indication of inherent fertility. | We have added "because they have not been subjected to nutrient removal through cropping or extensive soil erosion that is evident on degraded uplands" (line 103). |
| 106 | Ref asks for detail on size of field and the characteristics of transects used for sampling | The paper says that (i) toposequence length (over which sampling occurred) is typically 0.5-5 km (Fig.1), (ii) the research watersheds were typically <5 km$^2$ (line 93) and (iii) farmers typically have <1 ha land fragmented into small fields along the toposequence (line 77). We have added that the field sampling transect was 'z-shaped' (line 107). |
| 111 | Ref "It is not clear why the authors refer to unpublished data that refer to a site that is not under analysis in this work. If this set of data should substantially contribute to the understanding of this work, | No change made. At lines 67-69 of the Introduction we cite the work of Cornish et al (2010) in Purulia District to explain our focus on the analytes reported. Data on soil profile pH from that research were not published, but they are relevant to the present study in which we did not have the resources to study subsoil pH (now explained in line 112). Lines 113-114 provide details on sampling procedures, the analytical method is at line 118, and Cornish et al (2010) provide site descriptions. |

| | then should be clearly stated and explained in the text" | |
|---|---|---|
| 140 | The minimum values of the pH depicted in Fig 2a for non-rice and rice fields seems to be different when in the text the two type of soils have the same value, i.e. 4.4. | Text revised. In the original text, values were rounded to one decimal point (both 4.4), but the means were actually 4.35 (non-rice) and 4.44 (rice). The 0.09 difference evident in Fig. 2a is the actual difference.
We have amended the values on line 140-41 to 2 decimal places. |
| 148,152, 155 | Purulia is often mentioned in this work notwithstanding it is not amongst the District under analysis. Authors should clarify the role of this District in this work | See comment on line 111. |
| 164 | It would be interesting to know the value of "potential yield based on rainfall". This would contribute to better understand the discussion about rice productivity. | We are not aware of any research specifically on this in E India, and our own work is unpublished. We measured yield in hundreds of farm fields and in experiments, with around 7 t/ha as the maximum in both, although rarely in farm fields. Also, in our previous research, we estimated that annual ET of medium duration rice (125 days) in this region is around 460 mm. If we assume transpiration is around two-thirds of ET, then T is around 280 mm. If transpiration efficiency is 25 kg grain/ha/mm then a potential yield would be around 6.9 t/ha for this class of land and crop duration. This is an interesting point, but we are reluctant to cite our unpublished work. |
| 169 | Can authors specify why homestead land "being the destination for all harvested materials including crop residues" was expected to have a higher OC content? | We think this is self-evident from Fig. 7, but we have added a note referring readers to Section 4.2 for further explanation. |
| 172 and 210 | We think the referee is asking for box plots for CEC and K | These are already given in Fig 3c. and Fig 5b |
| 204 | "...may be less than a third of potential". Also in this case it would be useful to have some data | Minor rewording. We did not study crop responses to added P, so there are no data. Reference to potential yield here relate to experiments in other environments and values are not directly relevant. The point here is not the potential yield, but that soil P on the EIP is very, very low by all of the published benchmarks (4 references cited), so we can expect crops to be mostly P-limited and to be very responsive to applied P (line 208). |
| 236-45 | The referee asks for a "clear definition of what soil fertility is in this work and to which extent the different elements (OC, CEC, K, etc.) taken into consideration contribute to its setting up." | The referee is right that soil fertility may include a large number of variables. Those chosen for study depend on the aim. We were concerned with soil chemical fertility (as stated at line 68), and with indicators that our previous research had shown to be agronomically important (lines 67-68). The Introduction states that we aimed to provide a foundation for improving plant nutrition (line 70). We think this is sufficient 'definition' of fertility as applied to our study.
Lines 239-41 has been clarified and a note on boron added on line 319. |
| 248 | The referee asks for some key detail about the geology of the analysed EIP watersheds should be provided. | The broad geological description is in lines 80-85.
No specific observations were made of site geology, so we are unable to provide more detail. Watersheds were chosen to represent diverse geologies, based on descriptions found in 'Soil maps, Department of Agriculture, Animal Husbandry and Cooperative. http://agri.jharkhand.gov.in/default.asp?ulink=resources/soilmap.asp.' |
| 251-52 | The referee refers back to line 248 and queries the point being made | Now clarified in the text, lines 248-52. The point we are making is that our results were quite consistent across sites, unlike in other studies, and we attribute this to the relative homogeneity of watersheds, notwithstanding differences in geology. That is, although geologies varied, the fertility trends generally did not. |
| 257-59 | Referee suggests briefly explaining recommendations | We have provided references for readers requiring more detail (lines 259-60). We trust this is sufficient given the paper is primarily about soil fertility |

| | | |
|---|---|---|
| | and providing detail about the learning process. | not how farmers learn. Having concluded that site variability indicates the need for site-specific management (256), our discussion in lines 259-60 noted the challenges of implementing site-specific management in the socio-economic context of East India and suggested how these might be addressed using a participatory learning approach. |
| 266 | Clarification of animal carcases as nutrient sinks. | The category "Old cows burnt, buried or sold" represents a sink for nutrients because they don't contribute to nutrient flow within the farming system (in the relevant time-frame). Text has been added to clarify this (line 268) and also the possible sale of old animals in Fig 7. |
| 294 | Clarify tonnes of…? | Text has been amended to specify grain (line 296). |
| 293-310 | Requests revision of the discussion around limiting nutrients | Revised as requested. The referee is correct that these are approximate estimates – used to develop a hypothesis re. nutritional constraints (lines 299 and 300). Like any hypothesis, it requires testing (that will also improve the approximations). |
| 313, 345 | Sustainability definition | See response to general issues, above, and additions to Conclusions line 349. |
| 315 | Describe test strips etc | See response to comment on line 257-9 |
| 318-31 | Summarise the main fertilization techniques currently adopted by farmers. This would allow readers to better understand the discussion and the authors' recommendations. | Not clear why the referee requests this at this point in the paper, which is a section on soil acidity. Our discussion refers to future fertiliser practices that farmers might adopt to manage acidification, if necessary. Our only reference to current farmer practice is the widespread use on DAP at low rates on all crops (line 327-28), an observation we can't elaborate upon. |
| | | |
| | Referee 2 | |
| 2 | Title | Amended as suggested |
| 9 | Add 'wetland' | Done |
| 16 | Change wording to 'ameliorate' acidification | We prefer to keep the current wording. Management embraces a wider set of options than is implied by the word 'ameliorate', that typically means liming. Management may include, for example, the choice of N source and the timing of N application to minimise leaching. We used 'further' intentionally, because acidification has already occurred. |
| 146 | Comment on soil acidity | Text amended as suggested |
| 183 | Change wording | Text now reads "explaining much of the variation in CEC …" |
| 226 | Questions P/K relationship | The relationship is weak, as we say, but it *is* significant (P<0.05). It is the weakness that is actually important. We will add here that this was the only significant (but weak) relationship between any of the fertility indicators (line 226). We conclude the paper (line 356) by saying that weak associations between the fertility indicators highlights the need for field-specific fertiliser regimens |
| 252 | Suggest 'wetland' | Done |
| 254 | Suggest additional reference | The referee refers to a paper by Homma et al. (2003). We now cite this in line 29 (28 in the original MS), 66, 245 and 255. |
| 283 | Minor revision | Done line 285 |
| 299 | Question assumption of 100% efficiency for P fertiliser | We agree that efficiency will be less than 100%. Text has been revised to say 'even with 100% fertiliser efficiency' (l. 302) the yield attained would be only 2 t/ha, to support our argument that P is the primary nutritional constraint (l. 300). |
| 329 | Asks if there is evidence for micronutrient deficiencies | We had noted B deficiency in the original MS (line 124) and have now also noted in line 319 that B deficiency is widespread according to Anon (2013). |
| 346 | Suggest minor revision | Agreed, and text revised at line 350 |
| 349 | The referee notes that the constant removal of straw and grain from rice fields, and the leaching of nitrate N are | Agreed. This cautionary note has been added at lines 353-55. |

[revised manuscript text omitted]

---

## Author Response (AR2)

20th June, 2020

PO Box 82
Kurrajong NSW 2578
Australia
(Western Sydney University)

Prof John Quinton
Executive/Topical Editor

Dear John,

Re: Soil 09-12-19

Thank you for your kind words on out manuscript.

I have uploaded a new pdf version with an amended Fig 2b showing standard errors of the means. I've changed from a line graph to a bar graph, which I think is clearer.

I've also added 'regionally' in line 154, just before Fig. 2.

Kind regards

Peter

Peter S Cornish MScAgr, PhD
Emeritus Professor
Western Sydney University

Tel. +61 2 4573 1663
Mob. 0414244269

westernsydney.edu.au